# Adaptation and Validation of the Lithuanian Version of the Sport-Specific Doping Self-Regulatory Efficacy Scale

**DOI:** 10.3390/ijerph20054158

**Published:** 2023-02-25

**Authors:** Saulius Sukys, Beatrice Hoppen

**Affiliations:** Department of Physical and Social Education, Lithuanian Sports University, Sporto 6, LT-44221 Kaunas, Lithuania

**Keywords:** doping self-regulatory efficacy, validation and adaptation, doping in sport

## Abstract

Background: Use of banned performance enhancing substances in sport is one of the most widely recognized anti-doping rules violation. Research evidence suggests that self-regulatory efficiency is one of the key psychosocial processes related with doping. Therefore, aiming to generate more insights on the self-regulatory efficacy, sport-specific doping self-regulatory efficacy scale was proposed. The aim of the present study was to adapt and validate the Lithuanian version of the sport-specific doping self-regulatory efficacy scale. Material and Methods: The scale construct validity and reliability was tested using a sample of 453 athletes (mean age 20.37, SD = 2.29; 46% male). Structural validity was assessed by exploratory and confirmatory factor analyses, convergent and discriminant validity of the scale were evaluated by assessing average variance extracted and also via correlational analyses. Cronbach’s alpha and composite reliability values were used for reliability analysis. Results: Exploratory and confirmatory factor analyses confirmed the one factor structure of the sport-specific doping self-regulatory efficacy scale. The results also indicated that the scale had sufficient convergent and discriminant validity. The results showed an excellent level of internal consistency. Conclusions: This study makes a contribution by confirming the validity and reliability of the Lithuanian version of the sport-specific doping self-regulatory efficacy scale.

## 1. Introduction

According to the World Anti-Doping Agency (WADA), doping is defined as the occurrence of one or more of the anti-doping rule violations [1]. Although there are eleven rules, the most widely recognized anti-doping rule violation is an athlete’s use of a banned performance enhancing substance or method [2]. Daily cases of athletes using prohibited substances in sport continue to come up on news headlines as athletes want to gain competitive advantage in any way possible. Evidence of doping use among athletes shows that doping has become an important issue in sport. Some studies on the prevalence of doping in elite sports suggest that 14–39% of elite athletes are doping [3]. Although a survey by Ulrich et al. [4] showed that up to 57% of elite athletes use banned performance enhancing substances. According to WADA doping control tests, banned substances are used in various sports [5]. It is important to emphasize that use of doping is not only harmful and dangerous for the athletes’ health, but also threatens the moral integrity of sports [6]. Therefore, doping has become a big problem and one of the main challenges that must be overcome in the world of sports.

Understanding why some athletes dope is important if we want them to stop using the prohibited medications. Doping is influenced by various personal and social context factors, but a meta-analysis by Ntoumanis et al. [7] identified moral disengagement and self-regulatory efficiency as key psychosocial processes related with doping. Bandura’s [8] social cognitive theory of moral thought and action is used to explain the role of these psychological processes in athletes’ behavior. According to Bandura [8], individuals create moral standards that help regulate their behavior via evaluative self-reactions. Individuals feel good when their actions conform to their own moral standards. When individuals violate these standards through their actions, they feel bad. Therefore, individuals are more inclined to behave in such a way that actions will confer self-worth and satisfaction, and vice versa, they avoid actions that will cause self-condemnation [8]. This is related with emotional reactions. If certain actions are expected to cause negative emotions (such as guilt), this should discourage such actions. Therefore, if the use of doping is not justified from a moral point of view, it is likely that athletes who use it will feel guilty, and this should discourage them from such behavior. However, according to Bandura [8], individuals can reduce or completely eliminate the impact of negative emotional reactions by enacting a thought process termed as moral disengagement. Eight psychosocial mechanisms of moral disengagement were identified and by using any of these mechanisms individuals are able to disengage effective self-sanction from reprehensible behavior. It should be noted that only six mechanisms of moral justification have been identified in doping research [9,10]. Numerous studies have revealed that moral disengagement is related to doping likelihood and doping use [9,10,11,12,13,14,15].

Another important psychosocial process related with doping is self-regulatory efficacy. Self-efficacy has an effect on courses of actions directly and through its impact on motivational, cognitive, and effective factors. This can affect how well people motivate themselves in the face of trouble and distress. The self-regulatory mechanisms through which moral agency is exercised are important to the self-management of transgressive behavior [16] Thus, self-regulatory efficacy reflects the belief in one’s capabilities to resist personal and social pressures to engage in harmful behavior [16]. In this way, higher self-regulatory efficacy should lead to less frequent engagement in antisocial behavior [16]. In addition, it was observed that self-regulatory efficacy is negatively related to moral disengagement because individuals who firmly believe in their ability to resist the urge to behave badly have no need to justify and rationalize such behavior [16]. When focusing on doping, self-regulatory efficacy shows an athlete’s ability to resist both personal and social environmental factors influencing the use of prohibited drugs. Studies have provided evidence that doping self-regulatory efficacy has been associated with lower intention to use doping [10,17,18,19,20] and doping behavior [7,11]. Ultimately, it can be said that doping self-regulatory efficacy can discourage athletes from doping.

Aforementioned protective role of doping self-efficacy regulation encourages research aiming to better understand intention to use prohibited performance enhancing substances in sport. As the use of doping is a widespread problem [21] and can vary for several reasons, such as different laws, countries, or individual backgrounds [22], it is important to encourage such studies in various countries. This is particularly important in countries where such research is lacking. It must be noted that in Lithuania, one study examining moral disengagement was conducted [23]. Nevertheless, no empirical studies have been published examining doping self-regulatory efficacy among athletes. Hereby, when investigating doping self-regulatory efficacy, it is important to use valid measures. One of the research instruments used to assess doping self-regulatory efficacy is a 10-item scale developed by Lucidi et al. [10]. However, there are some concerns about this scale. First, it was based on interviews with adolescents. Second, almost half of the participants were not involved in sports, and it was not clear if they had any experience with doping. Finally, some items are related with physical appearance and are not relevant to sport performance. Recently, Ring and Kavussanu [20] adapted Lucidi et al. [10] scale and offered an abbreviated sport-specific version of the doping self-regulatory efficacy scale. This new version has both excellent factorial validity [20] and reliability [19,20]. Therefore, we believe that this version of doping self-regulatory efficacy scale has more potential in studies which examines athletes’ ability to resist doping. Hence, the adaptation of this scale to different languages is an important issue to facilitate cross-cultural comparisons.

In summary, we sought to adapt and validate the Lithuanian version of the sport-specific doping self-regulatory efficacy scale. Furthermore, we wanted to encourage future research and offer the possibility of including Lithuanian studies on doping self-regulatory efficacy in the international context.

## 2. Materials and Methods

### 2.1. Participants

The participants for this study were athletes from Lithuania. Recommendations for the sample size were taken into account when planning the study. Some authors recommended 100 participants as a minimum number for validation research [24]. However, other authors recommended 200 as a fair number, 300 as good number and 500 as very good number of participants for factor analysis [25]. Sample size can be also related to the number of variables, the number of factors, the number of variables per factor, and the size of communalities [26]. Since the sport-specific doping self-regulatory efficacy scale is a unidimensional measure having seven items, we defined a minimum number of participants as 200 in our study. Whereas to conduct exploratory factor analysis and confirmatory factor analysis with the same participants is not recommended [27], the sample size was set to at least 400 participants.

The final study sample consisted of 453 (208 females and 245 males) athletes with an average age of 20.37 years (SD = 2.29), which ranged from 16 to 29 years. Athletes represented a variety of individual (track-and-field, swimming, shooting, gymnastics; *n* = 283) and team (basketball, football, handball, rugby; *n* = 170) sports and had competed in their respective sport in average of 8.30 years (SD = 2.89). Among athletes, 30.0% (*n* = 136) were currently competing or had recently competed at the international level, 51.9% (*n* = 235) at the national, and 18.1% (*n* = 82) at the regional or university level. To access structural validity, the total sample of athletes was randomly split into two subsamples. Subsample one consisted of 226 (131 males and 95 females) and subsample two of 227 (114 males and 113 females) athletes.

### 2.2. Measures

In this study the sport-specific doping self-regulatory efficacy scale was used to adapt and validate this instrument for the Lithuanian language. For further validity analysis, athletes’ attitude toward doping, sport values, and likelihood to use doping were measured.

#### 2.2.1. The Sport-Specific Doping Self-Regulatory Efficacy

A sport-specific version [20] of the doping self-regulatory efficacy scale [10] is a self-report 7-item instrument. Participants were asked to indicate their confidence in their ability to avoid using banned substances to improve performance in sport in seven situations by using a 7-point Likert scale, ranging from 1 to 7, in which 1 is “not at all confident” and 7 is “completely confident”. The previous study has shown excellent internal consistency (α = 0.95) [20].

#### 2.2.2. Attitudes toward Doping

The 8-item Performance Enhancement Attitude Scale [28] was used to measure athletes’ attitude toward doping in this study. Participants were asked to indicate their level of agreement with each of the statements using Likert scale ranging from 1 (strongly disagree) to 6 (strongly agree). The Lithuanian version of this scale showed good internal consistency (α = 0.93) and validity [29]. The higher mean of eight item ratings shows more positive attitudes toward doping.

#### 2.2.3. Sport Values

Values were assessed using the Youth Sport Values Questionnaire-2 [30]. The questionnaire consisted of 13 items and measured moral (5 items), competence (4 items), and status (4 items) values. In this study, we used only subscales pertaining to moral and status values. Participants were asked to respond on each item by using 7-point Likert scale, ranging from extremely important to me (5), to the opposite of what I believe (−1). Mean scores reflected stronger values. The Lithuanian version of this questionnaire showed adequate internal consistency (ranging from 0.70 to 0.85) and validity [31].

#### 2.2.4. Likelihood to Use Doping

The participants’ likelihood to use doping was measured with two hypothetical scenarios which have also been used in other studies [12,29]. The first scenario described a situation where the participants had the opportunity to use a banned substance to enhance performance. The second scenario described a situation where the banned substance could be used to recover from injury. Participants had to answer on 7-point scale, ranging from 1 (not at all likely) to 7 (very likely).

### 2.3. Procedures

When this study was approved by the university ethics committee, it was translated following the recommended procedures [32]. As we used an adapted version of the doping self-regulatory efficacy scale, which was in English, forward translation by two independent translators from English to Lithuanian was conducted, and there was synthesis of the two translated versions. Back-translation by two independent translators from Lithuanian to the source language (English) was conducted and there was an evaluation of translated versions by two experts and final consensus. Finally, pilot testing with 35 athletes (20 male and 15 females) was conducted that helped to ensure that items are meaningful and clear to the target population.

The main study was conducted with athletes from various sports clubs across all of Lithuania. Prior to data collection, all athletes were informed about the aim of this study, voluntary participation in the study, and anonymity of the data. Participants were also informed that honesty in responses is very important. After consenting, they completed the questionnaire.

### 2.4. Data Analysis

Prior main data analysis, preliminary screening for missing values, outliers and normality was conducted. No missing values and extreme outliers were detected. The distribution of the data was evaluated by skewness and kurtosis. Items with values above two were considered as problematic [33]. For descriptive analysis means and standard deviations were calculated. The structural validity of the sport-specific doping self-regulatory efficacy scale was examined with Exploratory factor analyses (EFA) and Confirmatory factor analyses (CFA). Data were randomly divided into two subsamples. The dataset of the first subsample was used for EFA. Whether the data was suitable for EFA, the Kaiser-Meyer–Olkin (KMO) test and Bartlett’s test of sphericity were conducted. Data are considered suitable for factorability if KMO is greater than 0.60 and Bartlett‘s test is significant [34]. All variable loadings should be no less than 0.32, but loadings in excess of 0.71 are considered excellent [25]. In order to confirm the structure of the scale, CFA was conducted with the dataset of the second subsample. A one-factor model fit was examined using fit indices, such as the goodness of fit index (GFI), chi-squared test (χ2), root mean square error of approximation (RMSEA), confirmatory fit index (CFI), normed fit index (NFI), and Tucker-Lewis index (TLI). Acceptable model fit criteria were CFI and CFI > 0.90, NFI and TLI > 0.95, RMSEA < 0.08 [35]. Furthermore, the average variance extracted (AVE) was calculated in order to assess convergent validity of the scale. A value of AVE higher than 0.5 was considered acceptable [36]. Convergent validity as well as discriminant validity were also tested via correlation coefficients between the sport specific doping self-regulatory efficacy and sport values, attitudes toward doping, and likelihood to use doping scores. For the scale reliability analysis Cronbach’s alpha (α) and composite reliability (CR) values were calculated. All data were analyzed using IMB SPSS and AMOS programs.

## 3. Results

### 3.1. Descriptive and Item Analyses

The averages of responses and standard deviations for each of the seven items of the Lithuanian version of the sport-specific doping self-regulatory efficacy scale are presented in Table 1. As shown in Table 1, the items mean scores range from 5.93 to 6.09. The skewness values range from −1.07 to −1.43, whereas the kurtosis values range from 0.12 to 0.99 that is within the acceptable range.

### 3.2. Exploratory Factor Analysis

The factor structure of the 7 items of the Lithuanian sport-specific doping self-regulatory efficacy scale was examined using the EFA (principal component analysis). Sample adequacy was met (KMO = 0.933), and there was a significant correlation between the variables (χ^2^ (21) = 1650.344, *p* < 0.001). One factor emerged and explained 79.4% of the total variance. In addition, the line graph of the scale was examined (Figure 1). Table 2 provides an overview of the factor loadings. All item loadings were excellent and ranged from 0.83 to 0.93.

### 3.3. Confirmatory Factor Analysis

In a next step, CFA was used to test the one-dimensional structure of the scale determined by EFA. It was found that model showed an acceptable model fit, except critical RMSEA value (χ^2^ (14) = 68.3, *p* < 0.001), CFI = 0.97, TLI = 0.96, NFI = 0.96, RMSEA (0.06–0.12) = 0.09, 90% CI [0.06, 0.13]). The inspection of the modification indices suggested that it induced a correlated error from item 5 to item 7 (Figure 2). As both of these items were related in meaning, a respecified seven-item model increased RMSEA substantially (χ^2^ (13) = 19.8, *p* < 0.10), CFI = 0.99, TLI = 0.99, NFI = 0.99, RMSEA = 0.05, 90% CI [0.00, 0.08]).

### 3.4. Validity and Reliability Analysis

Firstly, the average variance extracted (AVE) was calculated to assess scale validity. It was found that AVE was 0.79. As this value was above 0.50, convergent validity of the scale was met. Then we examined the correlations among self-regulatory efficacy and other variables. It was found that self-regulatory efficacy was inversely associated with athletes’ attitude toward doping (r = −0.71, *p* < 0.01) and also status values (r = −0.56, *p* < 0.01), but positively related with moral values (r = 0.76, *p* < 0.01). Furthermore, self-regulatory efficacy was strongly negatively related to doping likelihood (r = 0.83, *p* < 0.01). The reliability of the sport-specific doping self-regulatory efficacy scale was analyzed using Cronbach’s α internal consistency coefficient and by calculating composite reliability value. It was found that this scale exhibited excellent internal consistency (α = 0.96). The findings on composite reliability also showed excellent reliability (CR = 0.96).

### 3.5. Gender and Age Differences

A one-way ANOVA showed that females’ (*M* = 5.98, *SD* = 1.23) and males’ (*M* = 6.01, *SD* = 1.23) scores on self-regulatory efficacy did not differ (*F* = 0.07, *p* = 0.79). Using median split, participants were divided into two groups by age (≤20 years of old (*n* = 189) and >20 years old (*n* = 264). Study results also did not reveal a significant effect regarding participants’ age on self-regulatory efficacy (*F* = 0.19, *p* = 0.67).

## 4. Discussion

Our study aimed to adapt and validate the Lithuanian version of sport-specific doping self-regulatory efficacy scale. The results of the study revealed that this scale can be used in further studies with Lithuanian speakers. At the same time, our research provided support that this scale could also be successfully adapted to other countries. This is important because in previous studies that examined doping self-regulatory efficacy, mostly English speakers were selected as participants [19,20], or English versions of the questionnaire were used, even if the subjects represented countries where English is not the official language [18].

Statistically, the Lithuanian version of the sport-specific doping self-regulatory efficacy scale yielded satisfying results on the validity and reliability of the scale. It is worth commenting that the sport-specific doping self-regulatory efficacy scale is a unidimensional instrument [20]. CFA results confirmed the one factor structure in a Lithuanian athletes’ sample with an acceptable fit using most of the criteria (GFI, CFI, NFI, TLI and RMSEA). Considering the reliability of the scale, the results showed an excellent level of internal consistency. Moreover, reliability was very similar to the original English version of the scale [19,20,37].

In this research, we also analyzed correlations among athletes doping self-regulatory efficiency and attitudes towards doping, sport values and likelihood to use doping. Many previous studies have found that athletes’ attitudes towards doping in sport are related to their intentions to use doping [10,22,29,38]. Therefore, it could be assumed that athletes who are more able to resist doping will also have more negative attitudes towards doping in sports. This assumption was confirmed by the obtained strong correlation between the mentioned research variables.

Research evidence revealed that sport values also are important for athletes’ likelihood to compete clean [39]. More specifically, recent studies found that moral values were positively related to athletes’ clean sport likelihood [40]. Moreover, even if other factors (e.g., use of sport supplements) could encourage athletes to think about doping, they probably would not start using it if they hold strong moral values [41]. Other studies also found that moral values were positively related to athletes’ self-regulatory efficacy, while on the other hand, status values were negatively related to athletes’ self-regulatory efficacy [42,43]. Therefore, it can be assumed that those athletes who hold stronger self-regulatory efficacy would also hold stronger moral values and lower status values. The results confirmed these assumptions because athletes’ self-regulatory efficacy was positively (strong effect) related to moral values and negatively (medium effect) related to status values.

A number of studies with adolescents and adult athletes of various competitive levels have provided evidence that doping self-regulatory efficacy is associated with lower likelihood to use doping [17,18,19,20,37,44,45]. Our study was no exception and found a negative correlation between doping self-regulatory efficacy and likelihood to use doping.

Analysis of gender differences showed that men and women did not differ in their doping self-regulatory efficacy. The results are partially similar to Petrou et al.’s [18] study, in which no relationship between gender and self-regulatory efficacy was found. Although in some studies gender differences in self-regulatory efficacy were found [10,22], these studies were conducted with adolescents. We also did not find age differences similar to Petrou et al.’s [18] study. It is worth noting that in many of the aforementioned studies on athletes’ doping self-regulatory efficacy, no data on gender and age differences were provided.

Several methodological implications and study limitations should be mentioned. Results on validity and reliability confirmed that the sport-specific doping self-regulatory efficacy scale could be successfully adapted in other non-Western countries. In general, almost only English speakers were tested with this scale [20,37,44,45]—only one study used the translated French version [43]. Therefore, our study showed that this scale could be easily adapted to another cultural context. However, it should be mentioned that our study did not analyze whether the adapted sport-specific doping self-regulatory efficacy scale predicted past or current doping behavior. Some previous studies have shown that doping self-regulatory efficacy was related to doping behavior [7,11]. However, at the same time researchers question how accurate data can be using self-report questionnaires [46,47]. Nevertheless, in order to find out how many athletes have used or are using prohibited substances and methods, scientific research uses both questionnaires [4] and data from doping control tests [5]. However, Ring and Kavussanu noted, “one of the difficulties in doping research is that doping is an illegitimate behavior, to which athletes are naturally reluctant to admit” [12] (p. 7). Therefore, asking athletes directly about whether they use doping, or not, may not be reliable for various reasons, such as self-protection [48]. To circumvent this issue, researchers measure variables such as doping intentions [10] or likelihood [17,18,19,20,37,44,45] as proxies for doping behavior. Mostly hypothetical situations are used in doping research to determine the doping likelihood of athletes [20,44,45], as we did in our study also. The future studies on adaptation and validation of this scale should consider include variables of doping likelihood. In addition, it should be taken into account that presented situations could be different. For example, benefits and costs situations can be associated with increased and decreased likelihood of doping [19]. Benefits situations refer to incentives that increase the likelihood of deciding to dope (e.g., monetary gain, encouragement from a coach or sports manager, low chance of being caught). Costs situations are related with deterrents that decrease the likelihood to use doping (e.g., risk of injuries, illness or the absence of explicit financial benefits). Some studies found direct negative effect of self-regulatory efficacy on doping likelihood in benefits situations, but no relationship in costs situations [45]. Despite the widely used hypothetical situations in doping research, future research on adaptation and validation of the sport-specific doping self-regulatory efficacy scale should also consider including doping behavior variables.

Adaptation and validation processes of instrument for another language should also include others variables and this implicate that we have to use additional research instruments. Previous studies revealed that doping self-regulatory efficacy is positively related with athletes’ anticipated guilt to use doping [20,44,49]. By adapting and validating sport-specific doping self-regulatory efficacy scale, it would also be useful to test relationship between the scores of efficacy and anticipated guilt. We did not test it because there are no instruments adapted to the Lithuanian language for the examination of anticipated guilt, which are widely used in research. However, future validation studies should consider testing the possible relationship.

## 5. Conclusions

The current study has provided support for the validity and reliability of the Lithuanian version of the sport-specific doping self-regulatory efficacy scale. The Lithuanian version includes seven items and appears to be an appropriate instrument for athletes to better understand the capabilities to resist the pressure of engaging in doping behavior.

## Figures and Tables

**Figure 1 ijerph-20-04158-f001:**
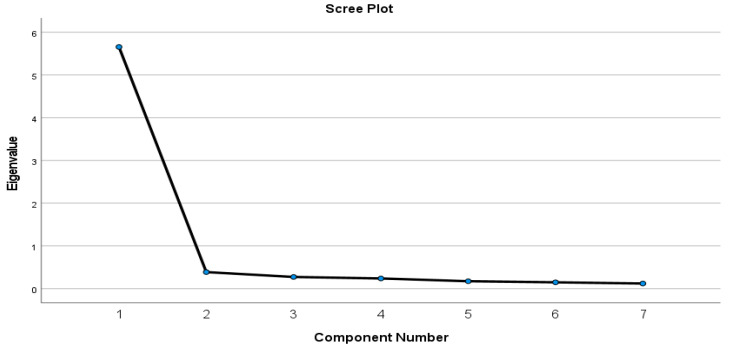
The scree plot of the sport-specific doping self-regulatory efficacy scale.

**Figure 2 ijerph-20-04158-f002:**
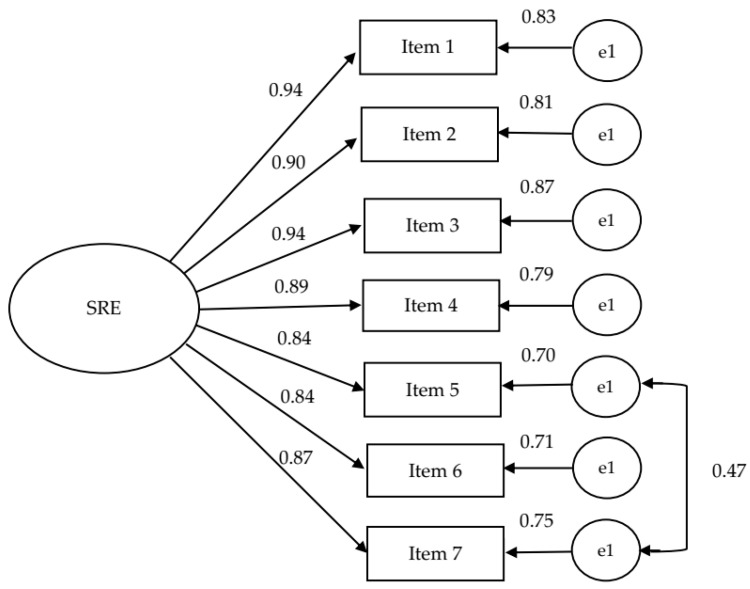
CFA of the sport-specific doping self-regulatory efficacy scale.

**Table 1 ijerph-20-04158-t001:** Descriptive and item analyses of Lithuanian version of the sport-specific doping self-regulatory efficacy scale.

	M	SD	Skewness	Kurtosis
1. *Jeigu dauguma sportininkų Jūsų sporto šakoje juos naudotų* [When most athletes in your sport use them]	6.04	1.26	−1.23	0.47
2. *Jeigu Jūsų fizinė forma suprastėtų (jaustumėtės nepakankamai pasirengęs)* [When you feel down physically (i.e., unfit)]	6.07	1.32	−1.43	0.99
3. *Jeigu Jums buvo liepta pagerinti savo sportinės veiklos rezultatus* [When you have been told to improve your performance]	6.09	1.28	−1.34	0.80
4. *Jeigu jaustumėte kitų spaudimą (pav., trenerio, vadybininko, rėmėjų) juos vartoti* [When pressured to do so by others (e.g., coach, manager, sponsor]	5.99	1.30	−1.07	0.12
5. *Norėdami pagerinti savo sportinės veiklos rezultatus, ir būdami tikri, jog draudžiami preparatai neturės jokio neigiamo šalutinio poveikio* [To improve your performance, even if it will not have any adverse side-effects]	5.98	1.31	−1.14	0.28
6. *Prieš svarbias varžybas, net kai galima būti nesugautam vartojus dopingą* [Before an important competition even when you can get away with it]	5.93	1.37	−1.25	0.96
7. *Norint greičiau pasiekti rezultatų, net jeigu apie draudžiamų preparatų vartojimą niekada niekas nesužinos* [To get results more quickly, even if no one would ever know]	6.02	1.39	−1.37	0.83

**Table 2 ijerph-20-04158-t002:** Items descriptive and factor loadings for the sport-specific doping self-regulatory efficacy scale in the EFA.

	Factor Loadings	Communalities
Item 1.	0.92	0.85
Item 2.	0.91	0.82
Item 3.	0.93	0.85
Item 4.	0.83	0.83
Item 5.	0.86	0.86
Item 6.	0.90	0.81
Item 7.	0.90	0.73

## Data Availability

The data presented in this study are available on request from the corresponding author.

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
