# Peer review of "Adaptation and Validation of the Lithuanian Version of the Sport-Specific Doping Self-Regulatory Efficacy Scale"

_ijerph, 2023, doi:10.3390/ijerph20054158_

Round 1

Reviewer 1 Report

Dear Authors

The study is interesting and highlights the use of scales to analyze the probability of the behavior of athletes in relation to the use of doping. However, I believe that the authors need to make some adjustments to improve the article and clear up some doubts.

I think the introduction is too long. I believe that the authors can reduce some parts of the text, agglutinating the common ideas in the same paragraph.

A key factor for the study is related to the characteristics of the sample participants.

Since the age of the participants is from 16 to 29 years old, I ask:

Are the participants professional athletes? How many?

Do participants participate in national and international competitions such as World Cups and Olympic Games?

In Lithuania, are the participants' individual and collective sports valued and recognized nationally and are they able to attract high financial investment?

In Lithuania, are these professional athletes well paid? Are athletes sponsored by the big brands in the sports industry?

How many of these athletes have already confirmed the use of doping?

How many of these athletes used doping and confirmed it through self-report?

Authors need to include this information in the article. This may influence the results of the study. Professional athletes may want to improve their sports performance in order to obtain medals and financial rewards. While younger athletes and or amateurs may have different goals from professional athletes.

I see that the authors should present these characteristics of the participants and explore these items in the discussion.

Authors should remove recommendations for future studies from the discussion and include in the “conclusions” section.

Sincerely

Reviewer 2 Report

1. Only 7 questions were investigated in this paper, each of which did not set the scene very carefully, so the workload was limited and the science was not enough.

2. There is no corresponding number and data to support the difference of age and gender, so I hope it can be made up.

3. In this study, there were no known cases of doping, and the sincerity of questionnaire filling was difficult to prove.

Reviewer 3 Report

An interesting and, unfortunately, still current topic in sports. Well written article. The authors clearly explained the purpose of the article, which they fulfilled. I have reservations about chapter 5, really brief, until it didn't have to be there. I recommend to work on it also in the context of the future solution of this issue.

Round 2

Reviewer 2 Report

no more questions.